# Biomass and Leaf Acclimations to Ultraviolet Solar Radiation in Juvenile Plants of *Coffea arabica* and *C. canephora*

**DOI:** 10.3390/plants10040640

**Published:** 2021-03-28

**Authors:** Wallace de Paula Bernado, Miroslava Rakocevic, Anne Reis Santos, Katherine Fraga Ruas, Danilo Força Baroni, Ana Cabrera Abraham, Saulo Pireda, Dhiego da Silva Oliveira, Maura Da Cunha, José Cochicho Ramalho, Eliemar Campostrini, Weverton Pereira Rodrigues

**Affiliations:** 1Setor de Fisiologia Vegetal, Laboratório de Melhoramento Genético Vegetal, Centro de Ciências e Tecnologias Agropecuárias, Universidade Estadual do Norte Fluminense, Avenida Alberto Lamego, 2000, Parque Califórnia, Campos dos Goytacazes, 28013-602 Rio de Janeiro, Brazil; wallace-bernardo@hotmail.com (W.d.P.B.); annersantos@outlook.com (A.R.S.); katherinefraga@yahoo.com.br (K.F.R.); baronidf@gmail.com (D.F.B.); cabrera.ana@hotmail.com (A.C.A.); 2Laboratório de Biologia Celular e Tecidual, Centro de Biociências e Biotecnologia, Universidade Estadual do Norte Fluminense–Darcy Ribeiro, Campos dos Goytacazes, 28013-602 RJ Rio de Janeiro, Brazil; saulopireda@hotmail.com (S.P.); diego_oliveira_3586@yahoo.com.br (D.d.S.O.); maurauenf@gmail.com (M.D.C.); 3PlantStress and Biodiversity Lab., Centro de Estudos Florestais (CEF), Instituto Superior de Agronomia (ISA), Universidade de Lisboa (ULisboa), Av. República, 2784-505 Oeiras, Portugal; cochichor@imail.telepac.pt or; 4Unidade de Geobiociências, Geoengenharias e Geotecnologias (GeoBioTec), Faculdade de Ciências Tecnologia (FCT), Universidade NOVA de Lisboa (UNL), 2829-516 Caparica, Portugal; 5Centro de Ciências Agrárias, Naturais e Letras, Universidade Estadual da Região Tocantina do Maranhão, Avenida Brejo do Pinto, S/N, 65975-000 Maranhão, Brazil

**Keywords:** fluorescence, leaf anatomy, leaf pigments, plant growth, UV-A, UV-B

## Abstract

Despite the negative impacts of increased ultraviolet radiation intensity on plants, these organisms continue to grow and produce under the increased environmental UV levels. We hypothesized that ambient UV intensity can generate acclimations in plant growth, leaf morphology, and photochemical functioning in modern genotypes of *Coffea arabica* and *C. canephora*. Coffee plants were cultivated for *ca.* six months in a mini greenhouse under either near ambient (UVam) or reduced (UVre) ultraviolet regimes. At the plant scale, *C. canephora* was substantially more impacted by UVam when compared to *C. arabica*, investing more carbon in all juvenile plant components than under UVre. When subjected to UVam, both species showed anatomic adjustments at the leaf scale, such as increases in stomatal density in *C. canephora*, at the abaxial and adaxial cuticles in both species, and abaxial epidermal thickening in *C. arabica*, although without apparent impact on the thickness of palisade and spongy parenchyma. Surprisingly, *C. arabica* showed more efficient energy dissipation mechanism under UVam than *C. canephora*. UVam promoted elevated protective carotenoid content and a greater use of energy through photochemistry in both species, as reflected in the photochemical quenching increases. This was associated with an altered chlorophyll *a/b* ratio (significantly only in *C. arabica*) that likely promoted a greater capability to light energy capture. Therefore, UV levels promoted different modifications between the two *Coffea* sp. regarding plant biomass production and leaf morphology, including a few photochemical differences between species, suggesting that modifications at plant and leaf scale acted as an acclimation response to actual UV intensity.

## 1. Introduction

Climate changes have important potential impacts on the structure, function, and diversity of terrestrial ecosystems and consequently, national economies. Estimates of stratospheric ozone depletion and associated changes in ultraviolet radiation levels (200–400 nm) suggest that solar radiation can be one of the most damaging stress factors for many crops [1,2]. Current estimates of the ultraviolet index (UVI), thirty years after the considerations proposed by the Montreal Protocol, show that the prohibition of substances that deplete the ozone layer is highly efficient for the recovery of stratospheric ozone [1]. However, without the protocol, UVI values at northern and southern latitudes less than 50° could be 10 to 20% higher during all seasons, similar to what happened in 2018, compared to those observed UVIs in the 90′s [3].

*Coffea arabica* L. and *C. canephora* Pierre ex A. Froehner grow in tropical regions under somewhat different conditions. These two species, which dominate coffee trade worldwide, differ in their evolutionarily environmental conditions. *C. arabica* is originally from the African tropical rainforests of Ethiopia, Kenya, and Sudan, and is found at high altitudes of 1600–2800 m, with an average annual temperature between 18 and 22 °C, and annual precipitation ranging from 1600 to 2000 mm [4]. On the other hand, *C. canephora* is originally from the lowland forest of the Congo River, extending to Central and Western Africa, at altitudes lower than 1200 m, average annual temperatures between 24 and 26 °C, and annual precipitation greater than 2000 mm [5]. Despite the well-described environment under which coffee species have evolved, information about the impacts of UV radiation on these two economically important species is lacking.

In Brazil, most of the elite coffee plants have been selected under high irradiance of full sunlight conditions [6]. Solar UV is characterized by high energy levels, with significant impacts on the biosphere, namely on morphological, physiological, and biochemical processes of plants [7,8,9,10]. Although some studies reported that coffee plants show physiological and metabolic plasticity regarding altered availability of light quantity and quality at the leaf [11,12], plant [13], and canopy scale [14], nothing is known about the effects of UV on coffee growth and physiological traits.

Solar radiation includes ultraviolet (UV) (200 to 400 nm), visible (400 to 700 nm), and infrared radiation (greater than 700 nm) [15]. UV can be sub-divided in three bands, classified as UV-A (315–400 nm), UV-B (280–315 nm), and UV-C (200–280 nm), which significantly differ regarding energy levels and their interaction with biological processes [2]. UV-C radiation is completely absorbed by the atmospheric gases present in the ozone layer. A small proportion of UV-B (less than 5%), together with UV-A radiation (between 10 to 100 times more than UV-B radiation), reaches the Earth’s surface, triggering responses at molecular, cellular, and whole plant levels.

Exposure to high levels of UV is reported to cause alterations in plant morphology, such as reduction in plant height, increased axillary branching, negative effect on biomass accumulation, and changes in resource allocation in *Vicia faba*, *Sorghum bicolor*, *Amaranthus tricolor*, and *Glycine max* [16,17,18]. High UV-A mediates plant growth, i.e., decreases biomass accumulation and increases biomass partitioning to shoots and leaves in *Cucumis sativus* [19]. Elevated UV-A intensity generates changes in leaf size and leaf anatomy [20], enhances the thickness of the palisade parenchyma and the abaxial epidermis, and reduces the spongy mesophyll and adaxial epidermis thickness [21]. Increases in UV-B radiation can lead to chlorosis and necrotic spots on the leaves [7,16,22]. Additionally, physiological modifications under elevated UV-B radiation were associated with stomatal density reduction and/or regulation of the stomatal opening, with the latter regarding the specific impact of high levels of UV-B radiation on guard cells control mechanisms [23]. High UV-B levels can promote deleterious impacts on the photosynthetic performance by promoting oxidative stress conditions that will affect photosynthetic pigments [9,24], proteins, and lipids, while significantly increasing grana disorganization [10,25,26]. Although both photosystems are affected by UV-B, the efficiency of photosystem II (PSII) is particularly impaired, mainly in the reactions coupled to the Mn-binding site of the water splitting complex, and in polypeptides D1 and D2 [10]. This induces an inefficient electron transfer [27], with losses of PSII functioning of up to 68% under elevated UV-B [28].

Despite the relevant information concerning the effects and responses of elevated UV intensities in various plant species, no information is available regarding coffee species. Considering the origin of coffee species from the deep forest understory, the hypothesis was that the current UV levels have already impacted these species, mainly *C. arabica*, related to the possible investment in protection mechanisms, which demands a significant amount of metabolic energy. In this sense, we suppose that the current UV intensities provoke alterations in responses at the plant/leaf scale, with possible differences between the two main cropped coffee species. The work aimed to study the responses of two plant species grown under two UV solar radiation regimes during the juvenile stage, addressing the following key questions: (1) Is near ambient UV radiation intensity already causing different acclimations in the two coffee species when compared to reduced UV? (2) Can a reduced UV radiation intensity enhance photochemical efficiency, change leaf anatomical traits, and affect coffee plant biomass partitioning? (3) Is *C. arabica* more sensitive to UV radiation than *C. canephora*?

## 2. Material and Methods

### 2.1. Experimental Site, Species Description, and Light Microclimate

The experiment was conducted at the State University of Northern Rio de Janeiro, Campos dos Goytacazes (21°44′47″ S and 41°18′24″ W, at 10 m altitude), Southeastern Brazil, using two of the most important coffee species in Brazil: *Coffea arabica* L. cv. Catuaí Amarelo IAC 62 and *C*. *canephora* Pierre ex. A. Froehner cv. lB1. On 2 June 2018 (tropical cold period), 120-day-old *C. arabica* and *C. canephora* seedlings produced from seeds and cuttings, respectively, were transplanted to 32-L pots (containing substrate composed of Oxisol and cattle manure, 2:1), which was considered as the beginning of the experiment and denominated as the first day after transplanting (DAT). At that time, plants had three pairs of leaves and average heights of 45 and 43 mm for *C*. *arabica* and *C*. *canephora*, respectively.

During the experiment, all plants were regularly watered. Agricultural practices of coffee plant cultivation, including fertilization and disease control were used according to the species demands.

Eight plants of each species were randomly distributed and grown under each of two distinct UV solar radiation conditions: (1) near ambient UV environment (UVam) inside the mini-greenhouse, with lateral walls and roof of corrugated glass, which excluded low levels of solar UV (16% UV-A and 0% UV-B), and (2) reduced UV levels (UVre), with *ca.* 70% UV-A and 90% UV-B solar radiation cut off. Plants were maintained for six months under these conditions before initiating measurements.

Photosynthetically active radiation (PAR, W m^−2^) in the two environments was recorded using a data logger (model 2000 Weather Stations, Spectrum Technologies, Plainfield, Illinois, USA). The UV radiation (W m^−2^) in the incident light was monitored with a spectroradiometer (OceanOptics model USB2000+, Dunedin, FL, USA), distinguishing UV-A (315–400 nm) from UV-B (280–315 nm). Measurements of PAR, UV-A, and UV-B were performed daily at nine points in each UV environment. All data were collected every 15 min from sensors positioned at the top of the coffee canopies. The average diurnal and maximum values were calculated for each month from June to December 2018.

### 2.2. Plant Growth Traits

Plants under the two UV treatments started to be observed after six months of growth under the two UV environments, with the most of measurements taken in the seventh month. The individual leaf expansion, expressing the dynamic of individual leaf elongation [29], was assessed through the elongation of the central vein of tagged young leaves (initial length = 20.7 ± 4 mm), emitted at the fourth plagiotropic branch counting from the top of the orthotropic plant axis of each plant (n = 8). Length measurements were taken at intervals of two days, from 172 to 192 DAT (20 November to 10 December), with a ruler until the leaf reached its final length. The tagged leaves, one per plant, were used for anatomical and stomatal measurements.

On the 204 DAT (22 December), plant height was measured from the stem base to the top apex with a graduated ruler. The basal diameter of the stem was determined using a digital caliper (Starret^®^ model 2001, Columbus, Georgia, USA). The number of leaves was counted, while total leaf area per plant was measured using a leaf area meter (Li-3100, Li-Cor, Lincoln, NE, USA). Finally, leaves, stems, and roots were separated, and plant material was dried in a forced-air oven at 70 °C for 72 h, to determine the leaf, stem, and root dry mass, and to calculate biomass partitioning (%). Specific leaf mass (SLM, g m^−2^) was obtained from 5 cm^2^ leaf discs dried at 70 °C during 72 h. Leaf discs were collected from the tagged leaves used for leaf expansion measurements.

### 2.3. Leaf Anatomy

On 203 DAT (21 December), leaf imprints from the abaxial leaf surface (from the tagged leaves used for some of the previously mentioned measurements) were observed using a light microscope. Three samples (0.050 mm^2^ each) per plant and treatment (n = 8) were observed from one field of view. Stomatal density (SD) was determined as previously described by [30].

On 204 DAT (22 December), leaf blade fragments were obtained from the tagged leaves (n = 5) fixed in a solution of 2.5% glutaraldehyde, 4% formaldehyde, and 0.05 M of sodium cacodylate buffer at pH 7.2. Thereafter, the material was post-fixed in 1% aqueous osmium tetroxide solution and 0.05 M sodium cacodylate buffer for 2 h and dehydrated in ascending series of acetone. After dehydration, the fragments were infiltrated with epoxy resin (Epon^®^). Finally, the samples were soaked in pure resin, placed in molds, and incubated in an oven at 60 °C for 48 h, for polymerization and block formation. Using an ultra-microtome (Reichert Ultracut S, Buffalo Grove, Illinois, USA) with a diamond knife (Diatome^®^, Hatfield, Pennsylvania, USA), semi-thin cuts, with section thicknesses between 0.60 and 0.70 μm, were obtained. The sections were stained with 1% Toluidine blue and 1% borax buffer for 1 min. Sections were mounted using Entellan^®^ (Merck, Kenilworth, NJ, USA) and observed under bright field microscopy (Axioplan ZEISS, Berlin, Germany).

Leaf tissue anatomical values were calculated from cross sections of the middle third of the leaf blade. The thickness of abaxial cuticle, adaxial cuticle, and epidermis were measured using a 40x objective. The thickness of palisade and spongy parenchyma was observed using a 20x objective. Leaves of five plants per treatment were used (n = 5), where 25 fields of view were examined for each repetition. The images obtained were processed and analyzed using Image Pro-Plus digital image processing software (Media Cybernetics, Inc., Rockville, Maryland, USA).

### 2.4. Photosynthetic Pigments Evaluation

Photosynthetic pigment content was evaluated at 200 DAT (18 December), by collecting one leaf (located in the previously emitted metamer than the ones used for leaf expansion measurements) at 1.00 p.m. Five leaf discs (*ca.* 28 mm^2^ each) were cut into fine strips and placed in a test tube containing 5 mL of dimethyl sulfoxide (DMSO) and incubated at 70 °C for 30 min in the dark [30]. After cooling the extract in the dark, the absorbance of a 3-mL aliquot was analyzed spectrophotometrically (700 Plus Femto, São Paulo, Brazil) at 480, 649, and 665 nm. Chlorophyll (Chl) *a* and *b*, as well as total carotenoid concentrations (µmol g^−1^ of dry mass), were determined according to [31].

Anthocyanin content was determined according to [32] using a methodology adapted for *Coffea* sp. from the same leaves referred to for Chl: Five leaf discs (each of *ca.* 28 mm^2^) were cut into fine strips and placed in a test tube containing 3 mL of methanol + hydrochloric acid (1%) and incubated at 8 °C for 24 h. The anthocyanin content (µmol g^−1^) was calculated according to [33,34].

### 2.5. Chlorophyll a Fluorescence

Fluorescence measurements were performed on light-adapted leaves, on the 201 DAT (19 December), during four diurnal periods (at 8.00 a.m., and 1.00, 3.00, and 5.00 p.m.). The third pair of leaves counted from the top of branches was used, localized at the plagiotropic axes emitted from the fourth orthotropic metamer that formed plagiotropic branches counting down from the top of the plant. Fluorescence yield changes were estimated using a pulse amplitude modulation (PAM) fluorometer MultispeQ V1.0 (PhotosynQ LLC, East Lansing, Michigan, USA). From these measurements, the various estimations were performed: Fraction of PSII centers that were “open” (q_L_), a parameter estimating the fraction of PSII centers in open states based on a lake model from the photosynthetic unit), and the estimate of the yield of energy dissipated through non-photochemical photoprotective processes (Y_(NPQ)_) [35]. Linear electron transport (LEF) was estimated from the equation: LEF = f(PAR)·Yϕ_II_, where f = 0.45, the factor that relates the absorption of PAR and the fraction of absorbed light that is transferred to PSII centers, where ϕ_II_ represents the effective quantum yield [36]. A series of transmission measurements was performed over a range of progressively increasing light intensities to increase the dynamic range of the results [36].

### 2.6. Statistical Analyses

The experiments were conducted in a completely randomized design, ideally with eight replicates (plants) for growth traits, pigment content, chlorophyll *a* fluorescence, and five replicates for anatomical analysis.

Generalized linear models (GLM) were used to estimate the effects of the UV environment (UVam and UVre), species (*C. arabica* and *C. canephora*), and their interactions on the response variables measured only at the end of experiment via two-way analysis of variance (ANOVA). In the absence of the UV environment with species interaction, means corresponding to each principal factor level were compared by the respective ANOVA F test. For cases in which interaction was significant, the means corresponding to each UV level were compared by F tests within each species. The chlorophyll *a* fluorescence variables measured during four of the diurnal periods (8.00 a.m., and 1.00, 3.00, and 5.00 p.m.) were submitted to a three-way ANOVA. All data were evaluated for homogeneity of variance among treatments (four combinations of UV environment and species) using the Bartlett test [37]. Models were compared by the likelihood ratio test and, when appropriate, reduced models were adopted. Least-squares means and respective statistical errors (S.E.) were estimated from the fitted GLMs, the S.E. derived from GLM ANOVA being a residual mean square error for each response variable. 

The variation of leaf elongation over time was represented by linear regression models. The effect of UV or species on these models was compared using time as a covariable, resulting in “mean effects” within species or within UV environments. 

All statistical analyses were performed using R software (R Core Team, 2020), employing the “nlme” [38] and “emmean” [39] packages.

## 3. Results

### 3.1. Light Microclimate

At the beginning of the experiment (June 2018), the average maximum diurnal irradiance values were 520 and 470 W m^−2^, decreasing to the lowest recorded values of 410 and 390 W m^−2^ in August, and increasing afterwards to a peak of *ca.* 900 and 790 W m^−2^ in December (200 DAT), for UVam and UVre conditions, respectively (Figure 1A,B). The similar monthly variation pattern over the experimental period was observed in average diurnal monthly PAR (Figure 1) and UV radiation values (Figure 2). The monthly average diurnal maximum UV-A values ranged between 14 and 20 W m^−2^ for UVam, and between 4 and 6 W m^−2^ for UVre, representing an approximately 70% reduction in the latter (Figure 2A,B). Regarding the monthly average diurnal maximum UV-B radiation, the values ranged between 0.6 and 1.4 W m^−2^ for UVam, and between 0.2 and 0.4 W m^−2^ for UVre, in the same period, representing approximately 67% and 70% reductions in the latter, respectively (Figure 2C,D).

### 3.2. Plant Scale: Morphology and Growth Traits

The UV regime under which the plants were grown did not cause significant impact on plant height and main stem diameter (Table 1), but *C. arabica* had increased height when compared to C. *canephora* in both environments, while the stem diameter in C. *arabica* under UVre was greater than in C. *canephora* under UVam.

The UVre environment resulted in an increase in the total number of leaves by 21% in *C. canephora*, and consequently, the total leaf area significantly only in this species (Table 1). Under UVam, the total number of leaves was significantly lower in *C. canephora* compared to *C. arabica*, while under UVre, the two species showed similar leaf numbers.

Under UVre *C. canephora* increased leaf (*ca.* 36%), stem (*ca.* 27%), and root (*ca.* 44%) biomass, while *C. arabica* only increased stem biomass (*ca.* 25%), in comparison to the respective UVam plants (Table 1). Interestingly, the UVre condition resulted in a significant reduction in leaf biomass in *C. arabica* (*ca.* 11%). In agreement, total biomass significantly increased by 40% in *C. canephora* under UVre compared to UVam, while *C. arabica* was irresponsive to UV treatment. *C. arabica* produced lower total biomass than *C. canephora* under UVre.

Under UVam, *C. arabica* showed a greater leaf mass allocation than *C. canephora* (42.5 vs. 37.4%), but both species showed similar biomass allocation values (*ca.* 36–37%) under UVre (Appendix A). The allocation in stems was greater in *C. arabica* than in *C. canephora*, regardless of UV condition. *C. arabica* allocated more carbon into stems under UVre when compared to UVam (29.6 vs. 26.2%). As regards the biomass allocation in roots, similar values were observed for each species (*C. arabica*: 31–34%; *C. canephora*: 40–41%), regardless of UV condition, although *C. canephora* showed greater investment in roots than *C. arabica*.

The exposure to UVre significantly decreased the SLM, similarly in both species, although *C. canephora* showed higher values than *C. arabica* for each UV treatment.

### 3.3. Leaf Scale: Tissue Thickness over the Vertical Cut and Leaf Elongation Rate

A reduction in UV level (UVre) significantly decreased the thickness of the abaxial (AbC) and adaxial (AdC) cuticle layer in both species (Table 1). No differences were observed between species for either UV treatment in both AbC and AdC, except for a greater AdC value in *C. arabica* than in *C. canephora* under UVam. AbC declined *ca.* 21 and 22% under UVre when compared to UVam for *C. arabica* and *C. canephora*, respectively.

The thickness of the abaxial epidermis decreased in *C. arabica* under UVre, while the thickness of the adaxial epidermis was mostly irresponsive to UV decline in both species (Table 1). Both *C. arabica* displayed greater thickness of the abaxial epidermis than *C. canephora* under UVam, and of the adaxial epidermis under UVre.

As regards the leaf palisade and spongy parenchyma thickness, no significant changes were caused by UVre in both species, as compared to their respective UVam values (Table 1, Figure 3). However, it was noteworthy that *C. arabica* showed a lower thickness than *C. canephora* in the palisade parenchyma, whereas the opposite was observed for the spongy parenchyma for both UV conditions.

The UVre environment decreased stomatal density in *C. canephora* when compared to the UVam environment (Table 1). A higher stomatal density was observed in *C. canephora* than in *C. arabica*, regardless of UV condition.

The main leaf vein elongation rate in *C. arabica* was on average 0.4318 and 0.5096 cm for the two-day intervals, attaining lengths of 10.68 and 11.64 cm at the end of linear elongation period, for UVam and UVre, respectively (Figure 4). In contrast, *C. canephora* showed a greater leaf elongation under UVam, with average elongation rates of 0.5148 and 0.4373 cm for the two-day intervals, attaining a length of 12.31 and 11.28 cm at the end of linear elongation period, for UVam and UVre, respectively.

### 3.4. Leaf Photochemical Responses: Photosynthetic Pigments and Chlorophyll a Fluorescence

The UV conditions did not significantly impact chlorophyll (Chl) *a*, Chl *b*, or total Chl content in the two species (Figure 5A–C). The Chl *a*/*b* ratio increased under UVre in *C. arabica*, but *C. canephora* was not impacted by UV conditions (Figure 5D). On the other hand, total carotenoid content significantly decreased under UVre in both species (Figure 5E). This implicated an increased tendency in the ratio of total Chl/total carotenoids under UVre in *C. arabica* (Figure 5F). The UV regimes did not have a significant impact on anthocyanin content (Appendix A), maintaining values between 0.002 and 0.003 µmol g^−1^, but *C. arabica* showed higher anthocyanin contents than *C. canephora* in UVam.

Linear electron transport (LEF), measured in light-adapted leaves, was not altered by UV radiation (Figure 6A). However, higher LEF values were obtained in *C. canephora*, maintaining greater values than *C. arabica* during all evaluated diurnal periods. LEF values were higher until 1.00 p.m., decreasing afterwards, for both species in both UV conditions.

The fraction of “open” PSII centers (q_L_) was reduced under UVre when compared to UVam similarly for both species (Figure 6B). Throughout the diurnal period, similar q_L_ values for each treatment were maintained until 3.00 p.m., increasing only at 5.00 p.m.

The ratio of energy dissipated through non-photochemical processes (Y_(NPQ)_) also decreased under UVre in both species, as compared to their respective UVam values, in all evaluated periods (Figure 6C). Notably, *C. arabica* maintained greater Y_(NPQ)_ values compared to *C. canephora* in both UV conditions and throughout the diurnal period, reflecting a higher energy dissipation through non-photochemical processes. Regarding the evaluation of the diurnal period, higher Y_(NPQ)_ values were observed in the period of 1.00 to 3.00 p.m. than at 8.00 a.m. and 5.00 p.m.

## 4. Discussion

The findings of this study offer the first integrated view at whole plant and leaf level of morphological, anatomic, and photochemical impacts of a reduction in UV-A and UV-B solar radiation on young plants of the two economically most important *Coffea* species.

### 4.1. Morphological and Anatomical Responses to UV Radiation

Morphological and anatomical responses in *Coffea* sp. supported our initial hypothesis that near ambient UV solar radiation intensity is provoking impacts that differed between the two coffee species. The increased participation of UV in ambient solar radiation negatively affects biomass accumulation of some species, such as soybean [40], sorghum [27], and wheat [41]. Furthermore, leaf expansion is one of the most sensitive growth parameters impacted by UV-B radiation [7]. The somewhat lower responsiveness to UVam at the plant and leaf scale in *C. arabica* was probably related to the species sites of origin, i.e., high altitudes of African tropical rainforests for *C. arabica* and large forest stands with altitudes lower than 1200 m for *C. canephora* [42]. High altitudes naturally receive greater levels of UV solar radiation when compared to the low altitude sites of origin of *C. canephora*, which could naturally select adaptations to UV solar radiation intensity in *C. arabica*. The modern *C. arabica* genotype used in our study has been selected for cultivation as a monoculture in full sunlight, which may also contribute to UV tolerance to some extent [6], supporting greater stability of this species regardless of UV conditions.

Despite similar main leaf vein elongation rates between environments for both species (Figure 4), UVre increased leaf area in *C. canephora* associated with greater number of leaves (Table 1). Leaf area determines light interception, thus, it is an important trait in determining crop growth and yield [43]. In fact, *C. canephora* had increased total biomass under UVre (Table 1). Therefore, reducing UV levels in the coffee canopy, especially for *C. canephora*, could be a potential strategy for increasing coffee yield.

Both species had higher SLM under UVam (Table 1), as also observed in several previously studied species such as soybean and cucumber [44,45], which could be related to the increased investment in mesophyll cells. Leaf cell number, dimension, and mass density determine SLM [46]. An increase in SLM related to increased leaf density might cause mesophyll cells to be densely packed [47], or to increase accumulation of metabolites [48], predominantly starch [45]. The SLM was significantly higher in *C. canephora* than in *C. arabica*. The increasing mass (leaf thickness) is favored in high altitude vegetation as a key strategy of high-altitude plants for efficient resource capture in harsh environments [49]. This is opposite to the differential altitude at the sites of origin of the two *Coffea* species studied here. It is worth noting that a species-dependent resource allocation was observed, regardless of the UV regime, with *C. arabica* displaying greater investment in the stem, while *C. canephora* had greater investment in the roots (Appendix A). However, *C. arabica* displayed lower leaf and greater stem investment under UVre, suggesting an acclimation at the leaf level and higher resource allocation in the stem.

Both coffee species had greater investment in both abaxial and adaxial cuticles under UVam compared to under UVre, which is likely to act as a protective mechanism in coffee leaves in relation to the UV solar radiation intensity. In fact, increased cuticle thickness provides protection against mechanical injuries and environmental changes [50,51], being considered the first barrier to high UV levels, especially for UV-A radiation [52]. This is associated with biochemical defense mechanisms, since cuticle tissue contains phenolic compounds, i.e., cinnamic acids, flavonoids, and flavones [53]. The cuticle also has a screening potential for UV radiation and an antioxidant capacity [54]. Additionally, as superficial tissues, the cuticle layers in adaxial and abaxial leaf surfaces act also as biophysical barriers by reflecting light, scattering, and reducing light absorption by epidermal layers [55]. Moreover, *C. arabica* displayed greater epidermal thickness than *C. canephora* under UVam, which could promote an improved ability to cope with higher UV radiation levels through epidermal transmittance and screening [56,57,58,59].

Our findings suggested that mesophyll thickness differed between the two coffee species. *C. canephora* displayed a thicker palisade parenchyma, whereas *C. arabica* displayed greater thickness of the spongy parenchyma, but both leaf tissues were irresponsive to altered UV conditions (Table 1, Figure 3). Phenolic synthesis in the leaves occurs in the mesophyll tissue and can have a substantial role in UV attenuation by scattering the short electromagnetic wavelengths by those molecules [60]. However, in this present research, modifications of UV levels did not show significant impact on anthocyanin content (Appendix A), but a greater content in *C. arabica* than in *C. canephora* supported the segregation in environmental adaptability of the two species.

Stomata play a crucial role in the control of leaf photosynthesis, regulating the precise balance between CO_2_ fixation and water loss to the atmosphere [61]. In this way, the balance of stomatal size and density is crucial to determine the diffusion of CO_2_ into the leaf. Interestingly, UV levels promoted changes in stomatal density (SD) in a species-dependent manner (Table 1). Under the two UV conditions, *C. arabica* did not show any difference in SD, whilst *C. canephora* had reduced SD under UVre. A genotype-dependent response to UV-B has been observed in rice [62] and soybean [63] with a greater reduction in SD on the adaxial surface than on the abaxial surface. Coffee leaves develop stomata only on the abaxial side, which is still an adaptation to excess of light [64]. Anatomic modifications, such as changes in SD, can modify stomatal conductance [65]. In this way, the anatomic changes at the stomatal level, together with those observed for both cuticle and epidermal thickness could, in turn, affect the leaf gas exchange dynamics [66].

### 4.2. Photochemical Responses to UV Radiation

The novel insight of our study was related to the total acclimation of *C. arabica* and *C. canephora* to UVam, based on their photochemical responses. Chl *a* and *b* contents were not impacted by UV levels, although the *a*/*b* chlorophyll ratio declined in *C. arabica* under UVam (Figure 5). This suggested that near ambient UV levels might affect the organization of the light harvesting complex (LHC), before having an impact on Chl content. In our study, near ambient UV levels provided a better adaptive advantage for *C. arabica* than for *C. canephora* leaves, indicated by the reduced *a*/*b* chlorophyll ratio. The synthesis of Chl *b* confers an advantage by stronger absorption of a wider range of light waves [67]. Chl *b* is synthesized from Chl *a*, and it is catabolized after it’s reconversion to Chl *a* [68]. Chl *b* levels are determined by the activity of the three enzymes participating in the chlorophyll cycle, namely, chlorophyllide *a* oxygenase, chlorophyll *b* reductase, and 7-hydroxymethyl-chlorophyll reductase, which are more resistant to proteolysis than those that determine the Chl *a* activity related to photochemistry [68].

The Chl *a/b* ratio modifications in *C. arabica* leaves suggested acclimation to ambient UV levels of this species, which occur as a general angiosperm adaptation to various light spectrum ranges [69]. In this context, the reduction of the Chl *a/b* ratio changes under UVam in *C. arabica* suggests that coffee leaves adaptively developed a rearrangement of chlorophylls in the LHCs to improve the efficiency of photosynthetically active radiation [69]. The *C. arabica* plants grown under UVam, likely as an acclimation response, were probably characterized by fewer PSII polypeptides, preferential loss of chloroplast proteins, and a deficiency in the Chl *a*/*b* LHC, as found in acclimation to high irradiance in this species [70]. Additionally, plants grown under UVam maintained a greater carotenoid content than under UVre in both species, reflecting a greater requirement for chlorophyll photoprotection from eventual photo-oxidative conditions triggered by higher UV levels [71,72]. This is in line with the greater non-photochemical energy dissipation (Y_(NPQ)_) in the UVam plants of both species, which reflected the presence of stronger photoprotective mechanism associated with energy dissipation [73], as compared to their UVre counterparts. The higher Y_(NPQ)_ values in UVam than in UVre plants were accompanied by greater q_L_, reflecting a more efficient photochemical energy use (q_L_), despite the lack of differences in LEF between UV conditions (Figure 6). Altogether, these results showed no strong photochemical differences between species, suggesting a total acclimation to UVam (with higher carotenoid content and Y_(NPQ)_) that allowed the plants to achieve even greater photochemical performance (q_L_) than the plants under UVre.

## 5. Conclusions

In the evolution of two economically important coffee species, from forest shade in their African centers of origin, to the monoculture cultivated under the full sunlight, various acclimations have developed to mitigate the possible damage caused by increased levels of UV solar radiation. Our study showed these acclimations at whole plant and leaf levels. Under UVam, both species increased SLA, carotenoid content, leaf abaxial, and adaxial cuticle thickness, q_L_ and Y_(NPQ)_, while decreasing leaf and stem dry mass and the Chl/carotenoid ratio. Despite some morphological and anatomical differences between species in response to UVam—such as: (i) reduced root and total biomass, number of leaves and leaf area, increased leaf elongation rate, and SD in *C. canephora*, and (ii) reduced biomass allocation in stems, leaf elongation rate, and Chl *a/b* ratio, with increased abaxial epidermis thickness in *C. arabica*—no species difference was observed in photochemistry. This suggested a total acclimation to UVam (with higher carotenoid content and Y_(NPQ)_) that allowed the plants to show an even greater photochemical performance (q_L_) than the plants under UVre. The interlinked responses demonstrated that: (i) the UVam levels can generate significant modifications in plant and leaf morphology in coffee plants, (ii) these changes acted as an acclimation mechanism to near ambient UV levels, resulting in protection of the plants and increased efficiency in energy dissipation, leaf functions, and biomass production, and (iii) interestingly, *C. canephora* seems to be somewhat more sensitive to UV radiation than *C. arabica*.

## Figures and Tables

**Figure 1 plants-10-00640-f001:**
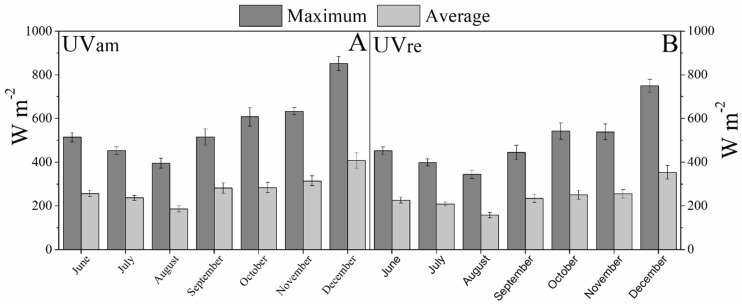
Diurnal maximum and average photosynthetic active radiation (PAR, W m^−2^) calculated each month, registered from June to December 2018 in either: (**A**) Near ambient (UVam) or (**B**) reduced (UVre) UV conditions.

**Figure 2 plants-10-00640-f002:**
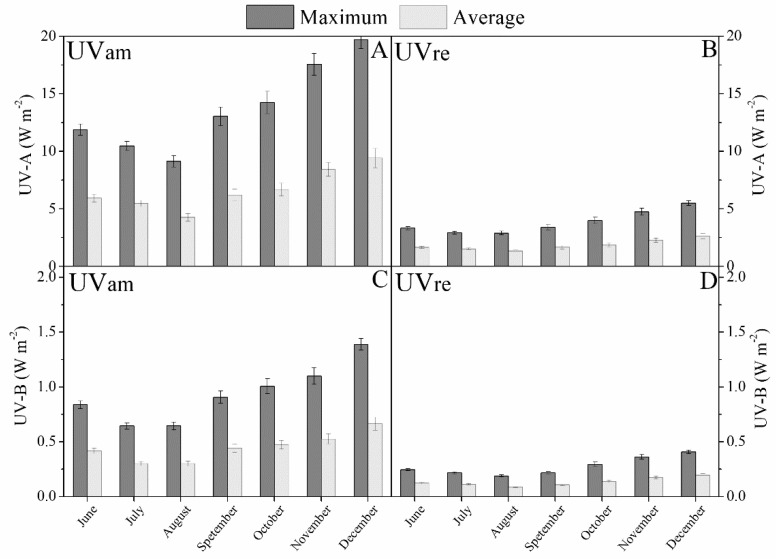
Diurnal maximum and average values of UV levels (W m^−2^) calculated for each month, and registered from June to December 2018. (**A**) UV-A in near ambient (UVam) environment, (**B**) UV-A in reduced (UVre) environment, (**C**) UV-B in UVam environment, and (**D**) UV-B in UVre environment.

**Figure 3 plants-10-00640-f003:**
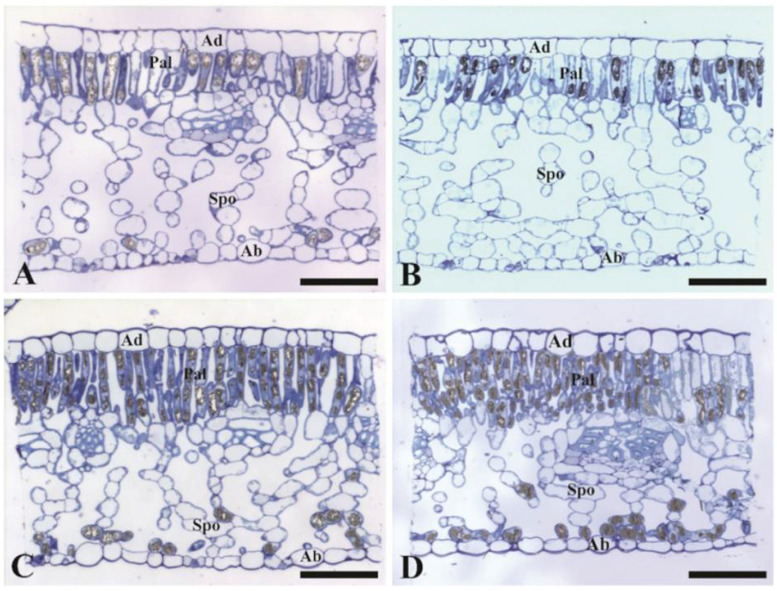
Anatomical aspects of the leaf blade of *Coffea arabica* (cv. Catuaí Amarelo IAC 62, (**A**,**B**) and *C. canephora* (cv. lB1, (**C**,**D**) grown under either reduced (UVre, (**B**,**D**) or near ambient (UVam, (**A**,**C**) UV radiations levels. The images are cross sections of the leaf blade observed through the light microscope. Ad—adaxial epidermis; Ab—abaxial epidermis; Pal—palisade parenchyma; Spo—Spongy parenchyma. Scale Bars: 100 µm.

**Figure 4 plants-10-00640-f004:**
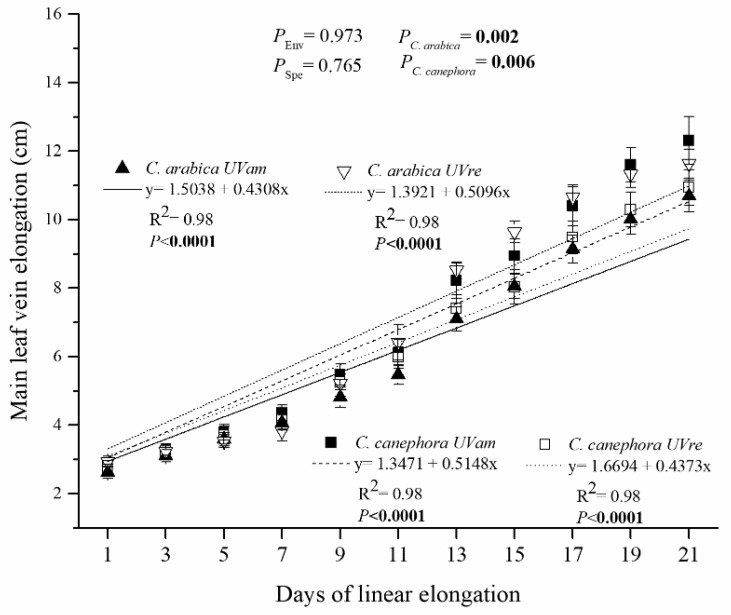
Main leaf vein elongation was measured between 172 and 192 DAT (days 1 and 21) for *Coffea arabica* and *C. canephora* grown under near ambient (UVam) and reduced (UVre) UV levels (Env). Estimated mean values ± S.E. (n = 8) and ANOVA *p*-Values are shown. ANOVA *p*-Values < 0.05 are marked in bold.

**Figure 5 plants-10-00640-f005:**
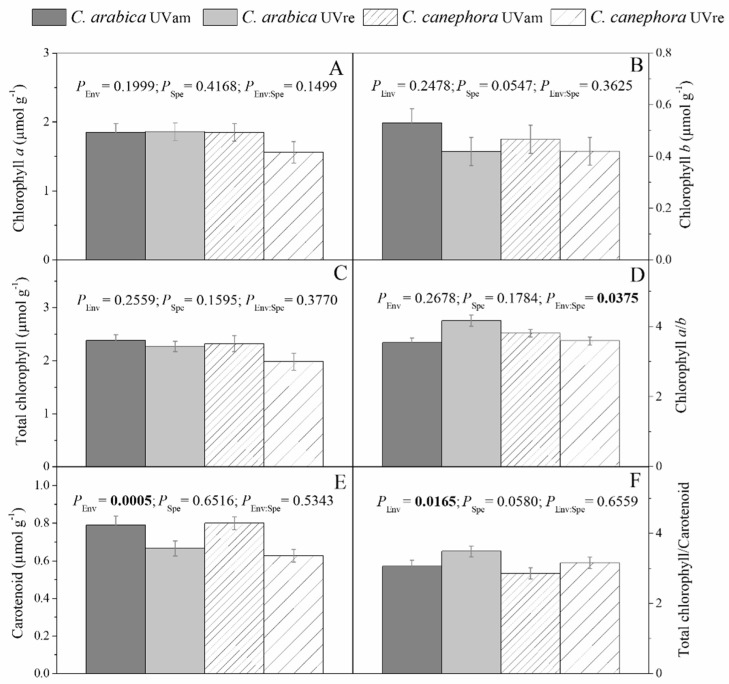
(**A**) Variations in contents of chlorophyll *a* (**A**), *b* (**B**), and total (**C**), chlorophyll *a*/*b* ratio (**D**), carotenoid content (**E**), and total chlorophyll (**F**)/carotenoid ratio for *Coffea arabica* and *C. canephora* grown under UV near ambient conditions (UVam) and reduced (UVre) levels (Env). The marginal significance was considered as 0.1. Estimated mean values ± S.E. (n = 8) and ANOVA *p*-Values for effects of species and UV regimes are shown. ANOVA *p*-Values < 0.05 are marked in bold.

**Figure 6 plants-10-00640-f006:**
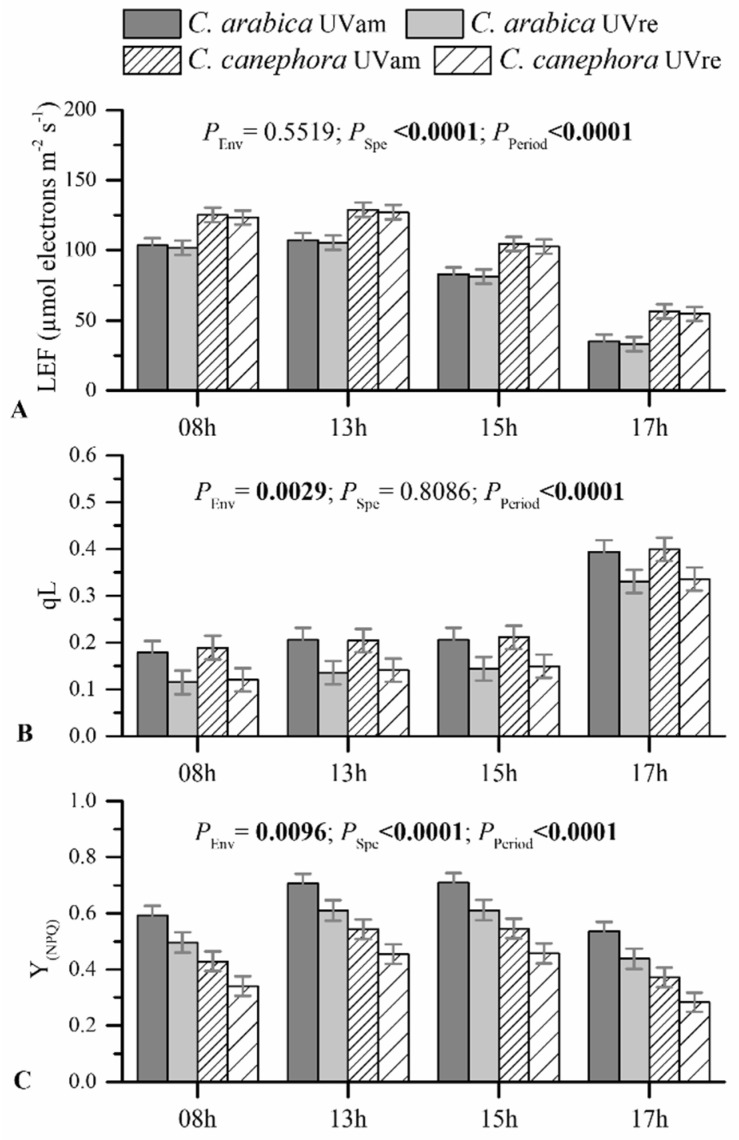
Linear electron transport (LEF) (**A**), the fraction of “open” PSII centers—q_L_ (**B**), and the yield for dissipation by downregulation—Y_(NPQ)_ (**C**), measured in light-adapted leaves during four diurnal periods (8.00 a.m., 1.00, 3.00, and 5.00 p.m.) for *Coffea arabica* and *C. canephora* grown under near ambient (UVam) and reduced (UVre) UV levels (Env). Estimated mean values ± S.E. (n = 8) and ANOVA *p*-Values are shown. ANOVA *p*-Values < 0.05 are marked in bold.

**Table 1 plants-10-00640-t001:** Plant growth traits and leaf anatomy parameters of *Coffea arabica* and *C. canephora* under near ambient (UVam) or reduced (UVre) ultraviolet levels. Mean values ± S.E. (n = 8 or 5) and ANOVA *p*-Values are shown.

Species	*C. arabica*		*C. canephora*		*p*-Value *		
Environment	UVam	UVre	UVam	UVre	Environment	Species	Environment: Species
Parameters	Plant Growth Traits						
Plant height (cm)	32.3 ± 1.52	32.4 ± 1.58	23.0 ± 1.52	23.1 ± 1.52	0.9280	**<0.0001**	0.7247
Basal stem diameter (mm)	12.6 ± 0.62	13.9 ± 0.65	11.4 ± 0.62	12.7 ± 0.62	0.0647	0.0723	0.3961
Total number of leaves	148.0 ± 5.58	153.0 ± 5.56	124.0 ± 5.58	156.0 ± 5.58	**0.0033**	0.0629	**0.0235**
Leaf area (m^2^)	0.58 ± 0.03	0.61 ± 0.03	0.47 ± 0.03	0.67 ± 0.02	**0.0006**	0.3345	**0.0149**
Leaf dry mass (g)	54.3 ± 2.45	47.8 ± 2.61	45.4 ± 2.45	61.6 ± 2.45	**0.0454**	0.4046	**0.0002**
Stem dry mass (g)	31.9 ± 1.65	39.9 ± 1.71	29.9 ± 1.65	37.9 ± 1.65	**<0.0001**	0.1925	0.2338
Root dry mass (g)	40.3 ± 4.61	43.70 ± 4.92	48.8 ± 4.61	70.1 ± 4.61	**0.0060**	**0.0008**	0.0520
Total dry mass (g)	128.2 ± 6.35	129.1 ± 6.35	122.5 ± 6.35	171.2 ± 6.35	0.9083	0.4965	**0.0005**
SLM (g m^−2^)	84.5 ± 2.96	74.7 ± 3.09	92.5 ± 2.96	82.6 ± 2.96	**0.0122**	**0.0303**	0.3791
	**Leaf anatomy**						
Abaxial cuticle (µm, 40×)	3.45 ± 0.17	2.72 ± 0.17	3.29 ± 0.17	2.56 ± 0.17	**0.0003**	0.2870	0.2917
Adaxial cuticle (µm, 40×)	4.31 ± 0.14	2.70 ± 0.14	3.73 ± 0.14	2.90 ± 0.14	**<0.0001**	0.2069	**0.0149**
Abaxial epidermis (µm, 40×)	17.2 ± 0.87	13.9 ± 0.87	13.1 ± 0.87	13.0 ± 0.87	**0.0173**	**0.0050**	0.0781
Adaxial epidermis (µm, 40×)	21.8 ± 0.94	21.3 ± 0. 94	20.3 ± 0. 94	19.8 ± 0. 94	0.5222	0.0548	0.1939
Palisade parenchyma (µm, 20×)	57.1 ± 2.97	55.6 ± 2.97	63.4 ± 2.97	66.0 ± 2.97	0.8523	**0.0162**	0.5138
Spongy parenchyma (µm, 20×)	164. 0 ± 7.12	161.0 ± 7.12	138.0 ± 7.12	135.0 ± 7.12	0.6692	**0.0049**	0.1013
Stomatal density (number mm^−2^)	192.0 ± 11.7	200.0 ± 12.5	320.0 ± 11.7	291.0 ± 11.7	0.5687	**<0.0001**	0.0884

ANOVA *p*-Values < 0.05 are marked in bold, whilst marginal values < 0.1 are underlined.

## Data Availability

Data is contained within the article or Appendix A.

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
