# Peer review of "Biomass and Leaf Acclimations to Ultraviolet Solar Radiation in Juvenile Plants of *Coffea arabica* and *C. canephora"

_plants, 2021, doi:10.3390/plants10040640_

Round 1

Reviewer 1 Report

Line number was strange.

Line 47, 201, 356, 364 ….. : What ca. means? There was no information about it.

Line 73-75 : Add reference about plant and UV light.

Line 83-95 : Add the importance of UV light in Coffea arabica L. and C. canephora.

Figure 2 : Note the scale bar means and explain the large scale bar in fig. 2 c on November.

Author Response

Comments and Suggestions for Authors

Line number was strange.

Answer:

We would like to thank the reviewer#1 for careful and thorough reading of this manuscript and for the thoughtful comments and constructive suggestions (We have accepted all suggestions), which help to improve the quality of this manuscript. It was solved as continuous and number of lines appear at right side of the text.

Line 47, 201, 356, 364 ….. : What ca. means? There was no information about it.

Answer:

“ca.” is the abbreviation for the Latin word “circa” that means “about” or “approximately”. It is widely used in scientific writing (as “e.g.” for “exempli gratia”) and there is no need of further explanation.

Line 73-75 : Add reference about plant and UV light.

Answer:

We have inserted the appropriate references.

Line 83-95 : Add the importance of UV light in Coffea arabica L. and C. canephora.

Answer:

Our study was exactly motivated by the lack of information about the impacts of UV radiation on Coffea sp. We have added such information into the text.

Figure 2 : Note the scale bar means and explain the large scale bar in fig. 2 c on November.

Answer:

Thank you for this observation. Someway, the statistical error was included with one decimal before the real value (lacking one zero)

Reviewer 2 Report

The research presented in the article relates to a very interesting and topical subject. However, the form of presenting the results requires considerable improvement.

  1. Authors' citations in the text and references should be adapted to the requirements of the editorial office.
  2. Affiliations should be written in English.
  3. The entire work is written in difficult and lengthy language. Please use shorter sentences and write briefly and to the point.
  4. The table includes dry mass of leaf, shoot, root. The authors, on the other hand, use biomass in the text, it is not the same.
  5. More detailed comments are provided in the text.

Author Response

Comments and Suggestions for Authors

The research presented in the article relates to a very interesting and topical subject. However, the form of presenting the results requires considerable improvement.

  1. Authors' citations in the text and references should be adapted to the requirements of the editorial office.

Answer:

We would like to thank the reviewer#2 for careful and thorough reading of this manuscript and for the thoughtful comments and constructive suggestions (We have accepted all suggestions), which help to improve the quality of this manuscript. To comply with the reviewer suggestion we have performed a thorough English language review. We modified the authors citation in the text and references section to conform to editorial office requirements.

  1. Affiliations should be written in English.

Answer:

Plants journal has published papers with Affiliation written in Portuguese (for example, Zinc Enrichment in two Contrasting Genotypes of Triticum aestivum L. Grains: Interactions between Edaphic Conditions and Foliar Fertilizers, Plants 2021, 10(2), 204; https://doi.org/10.3390/plants10020204). Therefore, we would like to maintain the Affiliation in Portuguese. 

  1. The entire work is written in difficult and lengthy language. Please use shorter sentences and write briefly and to the point.

Answer:

Thanks for pointing this. We have shorted some sentences.

  1. The table includes dry mass of leaf, shoot, root. The authors, on the other hand, use biomass in the text, it is not the same.

Answer:

Accoding to ScienceDirect (https://www.sciencedirect.com/topics/engineering/biomass), biomass is defined as matter originating from living plants, including tree stems, branches, leaves as well as residues from agricultural harvesting and processing of seeds or fruits. Additionally, “Biomass is often reported as a mass per unit area (g m−2 or Mg ha−1) and usually as dry weight (water removed by drying) (https://www.sciencedirect.com/topics/earth-and-planetary-sciences/biomass). Therefore, we inserted additional comments in the text to explain the use of biomass when needed.

  1. More detailed comments are provided in the text.

Answer:

Thanks for pointing this. We accepted almost all suggestions.

Answers related to the comments directly introduced in the manuscript by reviewer 2

(see peer-review-10986467.v1.pdf)

Line 197- I don't understand 'near ambient', either there is something in the environment or not. For me, near ambient means plants were near the UV waves, but the waves didn't touch him.

Answer:

We used near instead approximately, similar to other papers (for example, Solar UV‐B effects on PSII performance in Betula nana are influenced by PAR level and reduced by EDU: results of a 3‐year experiment in the High Arctic; https://doi.org/10.1111/j.1399-3054.2012.01596.x).

Line 2000 - I suppose that something is missing here

Answer:

Thanks for pointing this. We corrected the sentence.

Line 205 - photosynthetically active radiation (PAR)

Answer:

Thanks for pointing this.

Line 218 - plants began to be observed after six month as most of the measurements were taken after 200 DAT, which is almost 7 months

Answer:

We have improved such sentence.

Line 246 - What does mean a field of view

Answer:

The term "field of view" is typically used in the sense of a restriction to what is visible by microscope (please see  ‘Microscopy and Image Analysis’, Curr Protoc Hum Genet. 2005 August; 0 4: Unit–4.4. doi:10.1002/0471142905.hg0404s46.).

Line 280 - it means at 1.00 p.m. ?

Answer:

Yes. We have inserted the Reviewer’s suggestion.

Line 299- I don't understand this. It was at 5 p.m. or after  17th hours?

Answer:

Thanks for pointing this. We have improved the sentence.

Line 335 - please write time in English a.m. p.m.

Answer:

Thanks for pointing this.

Line 359 - please arrange the reference to the tables and figures in the text according to the editorial requirements

Answer:

Thanks for pointing this.

mass will be better word

Answer:

We already discussed this issue.

Please edit this passage, it is very difficult to understand. You can sometimes write that the plants had certain traits under a certain light, not just that the UV .. promoted etc. . Plants do not show anything, plants can be characterized, the results can show something

Answer:

We have improved this part of the text.

Please insert the English terms of hours, both in the text, in the figures and in the description of the figure. In this form it is confusing and it is not entirely clear.

Answer:

Thanks for pointing this.

maybe research or experiment

Answer:

Thanks for pointing this.

please use species not genotype in the whole text

Answer:

We have accepted the reviewer’s suggestion.

why surprisingly

Answer:

Several recent works have reported greater tolerant to abiotic stress in C. canephora than C. arabica. However, the latter surprisingly demonstrated better tolerance to actual UV radiation levels.